# Comparison of Micronutrient Intervention Strategies in Ghana and Benin to Cover Micronutrient Needs: Simulation of Bene-Fits and Risks in Women of Reproductive Age

**DOI:** 10.3390/nu13072286

**Published:** 2021-07-01

**Authors:** Mamta Dass, Jolene Nyako, Charles Tortoe, Nadia Fanou-Fogny, Eunice Nago, Joseph Hounhouigan, Jacques Berger, Frank Wieringa, Valerie Greffeuille

**Affiliations:** 1Institut de Recherche pour le Développement (IRD), BP 64501-911, av. d’Agropolis, 34394 Montpellier, France; dassmamta20@gmail.com (M.D.); jacques.berger@ird.fr (J.B.); franck.wieringa@ird.fr (F.W.); 2QualiSud, Univ Montpellier, Avignon Université, CIRAD, Institut Agro, IRD, Université de La Réunion, 34394 Montpellier, France; 3Nutrition Unit, Food Research Institute, Council for Scientific and Industrial Research (CSIR), Accra P.O. Box M20, Ghana; jolenenyako@yahoo.co.uk (J.N.); ctortoe@yahoo.co.uk (C.T.); 4Faculté des Sciences Agronomiques (FSA), Université d’Abomey-Calavi (UAC), Jéricho 03 BP 2819, Benin; nadia@fogny.bj (N.F.-F.); eun_nago@yahoo.fr (E.N.); joseph.hounhouigan@gmail.com (J.H.)

**Keywords:** micronutrients, fortification, supplementation, biofortification

## Abstract

Overlapping micronutrient interventions might increase the risk of excessive micronutrient intake, with potentially adverse health effects. To evaluate how strategies currently implemented in Benin and Ghana contribute to micronutrient intake in women of reproductive age (WRA), and to assess the risk for excess intakes, scenarios of basic rural and urban diets were built, and different on-going interventions were added. We estimated micronutrient intakes for all different scenarios. Four types of intervention were included in the scenarios: fortification, biofortification, supplementation and use of locally available nutrient-rich foods. Basic diets contributed poorly to daily micronutrient intake in WRA. Fortification of oil and salt were essential to reach daily requirements for vitamin A and iodine, while fortified flour contributed less. Biofortified products could make an important contribution to the coverage of vitamin A needs, while they were not sufficient to cover the needs of WRA. Iron and folic acid supplementation was a major contributor in the intake of iron and folate, but only in pregnant and lactating women. Risk of excess were found for three micronutrients (vitamin A, folic acid and niacin) in specific contexts, with excess only coming from voluntary fortified food, supplementation and the simultaneous overlap of several interventions. Better regulation and control of fortification and targeting of supplementation could avoid excess intakes.

## 1. Introduction

In West Africa, the prevalence of acute and chronic malnutrition is still high [1], but the magnitude of micronutrient deficiencies and their health impacts are poorly documented. In Ghana, a national micronutrient survey conducted in 2017 reported that deficiencies of iron, vitamin A and folate affect 13.7%, 1.5% and 53.8% of non-pregnant women, respectively [2]. In Benin, no national data are currently available on the micronutrient status of the population, but one study in women of reproductive age (WRA) reported a prevalence of 11.3% for iron deficiency and 17.7% for vitamin A deficiency [3]. Often, to address micronutrient deficiency, several approaches such as supplementation, fortification and/or biofortification and interventions to increase dietary diversity, are implemented simultaneously.

While micronutrients are essential nutrients for life, some of them can also be potentially harmful if consumed in excess. Most risk of excess intake comes with chronically high intakes, although, for some micronutrients, excess consumption can result in acute poisoning [4,5,6,7]. When consumed from natural food ingredients, micronutrients are generally supposed to be safe because of the regulation of their absorption in the body. However, alternative or pure chemical forms used for fortification or supplementation could have toxic effects when used in excess.

In several low- and middle-income countries (LMICs), different combinations of micronutrient interventions are currently being implemented and, while the overlap of these interventions could improve the coverage of micronutrient needs, it could also increase the risk of exposure to overconsumption for some micronutrients if not adequately monitored at the intervention level [8,9]. Recent studies have examined the potential risk of overconsumption due to the multiplication of interventions focused on vitamin A or iodine; these concluded that more studies are needed regarding the risk of overconsumption, as well as the dietary pattern of populations and a definition of UL [4,8,10].

The objectives of this study were to evaluate the following: How strategies currently implemented in two countries in West Africa (Benin and Ghana) contribute to the coverage of micronutrient needs in women of reproductive age; how existing strategies could be optimized; and whether there is a risk for excess intake of micronutrients targeted through these programs.

## 2. Materials and Methods

### 2.1. Micronutrients Programs

A situational analysis of nutritional programs currently or recently implemented, either voluntary or mandatory, in Benin and Ghana was performed as part of the INSIDER (INtegrated Strategies for MIcronutrient DEficiencies Reduction) project, funded by the European Commission under the Food Fortification Advisory Services. Fortification, biofortification and supplementation programs, whether mandatory, voluntary or currently in development, were documented through a desk review and interviews with local experts in Benin and Ghana.

In addition, the potential use of locally available micronutrient-rich foods such as fruit or fish was also considered, as they contribute to dietary diversification, a complementary nutrition-sensitive approach that can improve micronutrient intakes. Thus, alternative micronutrient-rich foods were included in the scenarios in replacement, or in addition to, basic diets, and their potential impact was estimated.

### 2.2. Target Populations

Two groups of women of reproductive age (15–49 years) were considered according to their different nutrient needs: non-pregnant and non-lactating women, and pregnant or lactating women. The scenarios were built for both rural and urban contexts, considering that the urban diet is more diversified, with women having access to a larger range of products than women living in rural areas.

### 2.3. Diet Patterns

Typical daily food consumption patterns were chosen as basic dietary patterns in order to calculate the basal amounts of micronutrients consumed. The number, type and amounts of meals were defined in consultation with local nutritionists in Benin and Ghana and according to results of previous published surveys [11,12,13,14,15,16]; personal communications from the Food Africa project dietary survey were also used [17]. Meal menus were designed based on popular, traditional meals routinely consumed by the population groups considered (Appendix A). Four dietary scenarios were constructed for women in both countries, with a more diverse diet for women in urban areas than women in rural areas, and supplementary dishes included in the daily meals of pregnant and lactating women. In Benin, the diet is mainly based on cassava and maize doughs accompanied by vegetable-based sauces. In Ghana, basic staples included in meals were rice, maize and yams. In both countries, wheat flour-based foods (bread and doughnuts) were considered to be available and consumed only in urban areas. A daily feeding pattern was defined as 6 meals per day for women in urban areas as follows: (1) breakfast, (2) morning meal, (3) lunch, (4) snack, (5) dinner, (6) after-dinner snack. For the rural population, the number of meals was set to 4, with the amount consumed per meal increased to fit the living habits of rural women.

Once dishes had been selected, recipes for each meal were determined based on published studies and reports [18,19,20]. The amounts of consumption (intake) for each food were estimated considering several data and surveys [11,21,22].

In addition to these basic diets, other sources of micronutrients from different interventions were included, and their participation in the coverage of micronutrient needs was calculated. The different combinations are described in Table 1.

The impact of the consumption of (bio)-fortified products on daily intake was estimated by replacing the basic micronutrient composition of the product with the micronutrient contents of a corresponding (bio)-fortified product. Therefore, basic maize porridge composition was replaced by one with biofortified maize. To include biofortified sweet potatoes, some of the snacks from usual diets (fried yam, orange, roasted groundnuts and watermelon) were replaced by fried sweet potatoes (Appendix A).

The impact of replacing basic oil with red palm oil in recipes, as well as the consumption of one mango (100 g) or the addition of fried tilapia (76 g) to a basic meal, was also tested. For pregnant or lactating women, consumption of fortified biscuits (voluntary fortification portion of 40 g) instead of fried tilapia was also tested. The chosen fortified product was a Lola milk biscuit, which belongs to the OBAASIMA scheme, which is coordinated by the Association of Ghana Industries and Ghana Standard Authority.

### 2.4. Micronutrients Intakes and Comparison to Recommendations

The total amount of micronutrient intake from diets was calculated by totaling the micronutrient content of each individual food ingredient reported in the FAO West African Food Composition Tables [23,24], or in the French Composition table Ciqual [25]. When available, the micronutrient composition of ingredients after cooking was used. The β-carotene contents in biofortified orange sweet potatoes and maize were estimated at 55 ppm (4.2 µg ER/g) and 6 ppm (0.5 µg ER/g), respectively [26,27].

The acceptable ranges for consumption were defined following the WHO/FAO recommendations [5] considering the estimated average requirement (EAR), corresponding to the average daily nutrient intake level that meets the needs of 50% of “healthy” individuals; the recommended nutrient intakes (RNI), corresponding to the EAR plus 2 standard deviations that meet the daily nutrient requirements for 97.5% of the population, and the upper limits (UL), corresponding to the maximum intake, are unlikely to pose a risk of adverse health effects from excess (Appendix A). For iodine, the upper limits indicated in tables were calculated according to WHO recommendations of 30 µg/kg/j for women with an assumed body weight of 55 kg and 40 µg/kg/j for pregnant or lactating women with an assumed body weight of 65 kg. For iron and zinc, the RNIs considered were those defined for a diet with low bioavailability, i.e., 5% for iron and 15% for zinc [5]. The WHO does not formulate recommendations for iron intakes by pregnant women but sets RNI for lactating women at 30 mg/day, a reference that was used for the whole group of pregnant/lactating women in the calculation of the coverage of iron needs. Since the WHO has not set a UL for iron, the UL proposed by IoM, which is 45 mg/day, was used [28]. The IoM UL is, however, below the RNI proposed by the WHO for diets with low iron bioavailability (5%) for women of reproductive age (58.8 mg/day).

## 3. Results

### 3.1. Micronutrient Interventions

Table 2 describes the ongoing micronutrient interventions implemented in Benin and Ghana. Three mandatory fortification programs are currently being implemented in both countries: salt with potassium iodate; oil with retinol palmitate; wheat flour with iron, folic acid and vitamin B complex. Mandatory salt fortification started in 1996 in Benin, 10 years before Ghana, and vegetable oil and wheat flour fortification has been mandatory since 2010 in Ghana and since 2012 in Benin.

Biofortification programs are currently under development in Benin. Three biofortified crops are currently being tested but have not yet been released onto the market: biofortified maize, biofortified cassava (both with vitamin A) and biofortified pearl millet with iron [29]. In Ghana, orange sweet potatoes and maize, both biofortified with Vitamin A, are already available, while biofortified cassava and pearl millet are still under evaluation.

Regarding supplementation in both countries, the Iron-Folic Acid (IFA) program provides iron and folic acid supplementation during prenatal consultation. In Ghana, the IFA supplementation program is also addressed to women with a child aged between 1 and 5 months.

### 3.2. Micronutrient Total Intakes

#### 3.2.1. Non-Pregnant, Non-Lactating Women

Table 3 summarizes the daily amount of micronutrients consumed from different dietary sources and the corresponding percentage of the recommended daily allowance (RNI), estimated average requirements (EAR) and upper limits (UL) covered in both urban (U) and rural (R) areas. Zinc was the only micronutrient for which the RNI was achieved through basal diet, and only for the urban scenario in Benin. Consumption of fortified food significantly improved the coverage of micronutrient needs, except for iron. In urban settings, the RNI of all micronutrients targeted by fortification—except iron—were reached in Benin, whereas, in Ghana, due to a lower amounts of wheat flour consumed, only EAR was reached for niacin, folate and iodine, and RNI was reached for vitamin A and zinc. In rural Benin, the lack of consumption of wheat flour impaired the coverage of micronutrient needs, and RNI was only reached for zinc, vitamin A and iodine. In rural Ghana, RNI for zinc was exceeded, and RNI for vitamin A and niacin was almost reached.

In Ghana, the consumption of vitamin A-biofortified products allowed for the reaching of the RNI for vitamin A. Furthermore, its combination with mandatory fortification programs resulted in a higher coverage of RNI, notably in rural settings.

In addition to fortification programs, the inclusion of grilled tilapia and mangoes in diets did not significantly change the coverage of RNI in Benin. In Ghana, it helped women reach the RNI for niacin and the EAR for folates in urban settings. In Benin, the replacement of vegetable oil with red palm oil in basic diets induced an excess intake of vitamin A that reached two times the UL. In Ghana, this replacement allowed participants to attain the RNI without reaching the UL.

#### 3.2.2. Pregnant and Lactating Women

Pregnant and lactating women have higher micronutrient needs than non-pregnant and non-lactating women (Appendix A). The basal diet only covered the RNI for zinc in urban settings in Benin, and only covered the vitamin A RNI in rural settings in both countries (Table 4). Considering other micronutrients, even EARs were not reached. In the scenarios drawn for Benin, the consumption of fortified products allowed participants to meet the EAR for niacin and the RNI for all the other micronutrients, except iron, in the urban setting. In the scenarios in Ghana, the lower consumption of wheat flour meant that the EARs for iron, niacin and folate were not met. In rural settings, only vitamin A RNI was reached in Benin and Ghana, and iodine RNI was only met in Benin.

The combination of basal diet with both mandatory fortification and supplementation resulted in most of the target population attaining the RNI for iron, zinc, vitamin A and folate in Ghanaian urban settings. In urban Benin, the RNI for these micronutrients, as well as iodine, was reached. In rural areas in Benin, the RNIs were not reached for zinc, niacin and folate; similarly, the RNI for iodine were not attained in Ghana. For areas in urban Benin exposed to a combination of interventions, intakes of folate and iron exceeded the UL. A similar scenario was observed in Ghana, where the addition of high-dose vitamin A supplementation, which was administered within the 2 months after delivery, resulted in exceeding the UL for vitamin A; this dose was 20 times higher than the UL.

In rural Benin, the intake of MN-rich foods in the form of fortified biscuits and fruits resulted in the coverage of all RNIs for all the major micronutrients, with the exception of zinc. Furthermore, ULs were attained in Benin for iron, niacin and folate, and in Ghana for iron, vitamin A (urban setting), niacin and folate. An intake of 40 g of fortified biscuits provided the RNI for iron and vitamin A, and the UL threshold for niacin, and exceeded UL for folate considering that 40 g of biscuit contains 1321.80 µg of folate while the UL is only of 1000 µg. When all interventions were combined with red palm oil consumption instead of fortified oil, the same coverage of needs was attained, but RNI and UL were reached for vitamin A in both Benin and Ghana.

## 4. Discussion

The present analysis of micronutrient intakes from common basic diets of women in two West African countries shows that strategies to increase micronutrient intake are urgently needed, as basic diets fail to reach the RNIs, or even the EARs, for most micronutrients. Indeed, combining different interventions aimed at improving micronutrient status would allow the RNI for pregnant or lactating women to be reached, albeit with some risks for excess micronutrient intake. Due to a different set of interventions, RNIs were only met for some micronutrients for non-pregnant, non-lactating women of reproductive age. Moreover, gaps in micronutrient coverage were higher in rural areas, as access to affordable, diverse foods and, notably, voluntary fortified food products, is much lower in rural settings in Benin and Ghana.

### 4.1. Role of Fortification in Coverage and Excess Intakes

Our analysis shows the essential role of food fortification in ensuring that WRA in both countries reach the RNIs for the micronutrients assessed, but it also stressed the need for adequate education and sensitization on fortification levels to ensure that consumption does not exceed UL.

In contrast to Benin, the combination of lower salt consumption and lower fortification level with iodine in Ghana resulted in insufficient intakes of iodine. Our simulations of micronutrient intakes were based on the hypothesis that products were adequately fortified. The 2017–2018 Demographic and Health Survey (DHS) in Benin found that 89.6% of households used iodized salt without measuring the level of enrichment [30]. However, in Ghana, while 65.5% of household had iodized salt, only 38.5% had salt with 15 ppm of iodine or more, according to the 2014 DHS [31], suggesting that iodine intakes are even lower in Ghana than our models show. Indeed, the 2015 national Iodine Survey found that WRA had a normal median urinary iodine concentration of 201.6 µg/L [32]. A recent study in children from Northern Ghana has also shown that bouillon cube containing iodized salt provided more than two-thirds of dietary iodine, stressing the importance of fortified food to the contribution of iodine intakes [33].

The fortification of oil with vitamin A also contributed significantly to women’s vitamin A needs. The higher level of consumption of fortified oil in Benin resulted in a higher contribution of fortified oil to daily vitamin A intake compared to Ghana. The low level of oil consumption in Ghana is in accordance with results of the 2017 National Micronutrient survey [2] that showed that, overall, 26% of households did not use fortified oil, with up to 83% of households in the Upper Western region not using vitamin A-fortified oil. In addition, only 56% of oil was adequately fortified, with great variations again apparent among regions [2]. These constraints (low usage, inadequate fortification level as well as the need for advocacy around oil consumption) are likely to impair the impact of the vitamin A oil fortification program in Ghana.

In contrast to salt and oil, the fortification of wheat flour with multi-micronutrients did not significantly contribute to micronutrient intakes due to the low level of wheat flour consumption in both countries; consumption of other cereals is preferred to wheat. In Benin, maize and rice are generally consumed more than wheat [11]. In Ghana, in a cross-sectional survey in urban area, bread made from wheat was found to be consumed daily by 62% of the population surveyed compared to maize, which was consumed daily by 96.8% of the respondents. These findings are in accordance with results of a recent study on the contribution of fortified food to the coverage of micronutrient needs in four African countries that shows that fortified foods are major contributor to vitamin A and iodine coverage, and poor contributors to iron coverage [22]. More popular food vehicles, such as maize or cassava flours, or popular condiments should be considered for fortification to achieve a higher coverage of micronutrients other than vitamin A and iodine.

Finally, to avoid exceeding the UL, voluntary fortification of products requires proper adequate political will, effective legislature and continuous regulation to ensure optimal coverage and increased effectiveness. For example, the brand of fortified biscuits chosen for the present study contained levels of niacin, folate and iron that were close to or over the UL set by the IoM, at least when the biscuits were used according to their instructions. The consumption of one portion (40 g) of biscuits, combined with the other interventions in place in the countries, resulted in iron and folate intakes >4 and >2 times the UL, respectively. However, the fortified biscuits were the only intervention that allowed the coverage iron needs of WRA, as neither the basic diet nor the mandatory fortification fulfilled daily iron requirements. Thus, while voluntary-fortified food could greatly contribute to the coverage of the needs of several micronutrients, clear legislation and guidelines should be put in place to limit the risk of overconsumption.

### 4.2. Biofortification

In Ghana, vitamin A-biofortified orange sweet potatoes and maize are already available while, in Benin, biofortified products are still being tested. In our scenarios, biofortified products made a significant contribution to the coverage of vitamin A needs, but this method would not be sufficient as a stand-alone to cover the overall micronutrient needs of WRA. To increase the contribution of biofortified products, efforts are needed to increase the acceptability of these products by consumers and producers, an issue raised by several studies regarding the marketing of biofortified foods [34,35]; that said, acceptability appears to be sufficient in some settings [36].

### 4.3. Supplementation

Supplementation programs for women are mainly focused on three micronutrients for pregnant or post-partum women: combined iron-folic acid supplementation (IFA) and high-dose vitamin A.

In the context of low amounts of iron and folate in the basic diet, and the low consumption of multi-micronutrient-fortified wheat flour, IFA supplementation is a major contributor to the coverage of iron and folate needs. Considering that the UL fixed by IoM is 45 mg of iron, the iron dose in standard IFA tablets (60 mg) exceeds the UL as a standalone intervention. Moreover, IFA supplements have been used by millions of pregnant women without recorded major side-effects. The UL for folate was not reached with the sole combination of fortification, biofortification or supplementation, but it was exceeded by the consumption of voluntary-fortified biscuits. Of note, in Benin and Ghana, as in many other countries, IFA supplementation targets pregnant or lactating women, and it is usually delivered through antenatal care visits. Non-pregnant, non-lactating women of reproductive age are not targeted by these programs. This study shows that several micronutrients are still deficient in the diet of women of reproductive age, even when they consume fortified and biofortified foods, especially for women living in rural areas. The consumption of locally available micronutrient rich foods could help to increase micronutrient intake, but their intake was not sufficient to cover the needs in our scenarios. Ensuring a good nutritional status prior to conception should be a public health necessity, and more nutrition programs should target adolescent girls outside the pregnancy and lactating periods.

The doses of vitamin A administered post-partum are far beyond the UL of 3000 µg RAE/day, as defined by the WHO, IoM or the European Food Safety Authority for preformed vitamin A forms (retinyl ester and retinol) [5,6,28]. Post-partum vitamin A supplementation was shown to have no beneficial impact on maternal mortality and morbidity [37], nor to improve the vitamin A status of a newborn at 6 months of age; in a systematic review and meta-analysis of seven trials, no evidence for a mortality or morbidity benefit was observed in infants [38]. Additionally, in Ghana, low-dose vitamin A supplementation administered weekly was found to be of no benefit in reducing mortality in women of childbearing age [39]. The WHO retracted this recommendation 10 years ago [40,41] yet, in many low- and middle-income countries, this program is still being implemented. In Ghana, according to the last DHS in 2014, 68% of women aged 15–49 with a child born in the past 5 years received a vitamin A dose post-partum [31]. Nevertheless, in the 2017 micronutrient survey, this proportion had greatly decreased, with “only” 23% reporting to have had post-partum vitamin A supplementation, which is still a considerable number for a non-recommended intervention [2]. Thus, considering the potential toxicity of high-dose vitamin A and the lack of efficacy of post-partum VAS, the implementation of this intervention is questionable, especially in a context in which alternative interventions already exist, as described in this study.

### 4.4. Micronutrients at Risk for Overconsumption versus Deficiency

Evaluation of risk of excess is challenging for several reasons. Firstly, ULs were defined for populations without deficiencies; whether they adapted to micronutrient-deficient populations is questionable, and highlights the need to consider the balance between the risk of deficiency and the risk of excess. Secondly, micronutrient toxicity is based on usual or chronic intakes rather than punctual high doses, as received in some supplementation interventions, such as high-dose vitamin A supplements. Below, we discuss the risk of micronutrient overconsumption in our scenarios for iron, vitamin A, niacin and folate.

Due to its crucial role in health and survival, vitamin A has been the subject of many interventions globally. In Ghana, during pregnancy or the lactation period, a woman could be offered two mandatory programs of fortification with vitamin A and two programs with vitamin A biofortification. Moreover, during the immediate post-partum period, she could receive a high-dose vitamin A supplement. In addition, she could consume different voluntary-fortified products. However, the combination and overlap of fortification, biofortification and MN-rich food does not result in a risk of overconsumption in the scenarios designed for Ghana, except when red palm oil was consumed regularly, contributing to 106% of the UL. In Benin, oil is currently the only food fortified with vitamin A, and no risk of overload was found in the scenarios. The replacement of vegetable oil with red palm oil created high amounts of carotenoids that largely exceeded the UL for vitamin A (two times the UL). However, the UL for vitamin A only includes preformed forms, such as those used for fortification or supplementation, whereas carotenoids, as found in red palm oil, are not reported to be toxic because of the downregulation of cleavage enzymes during high preformed VA intakes [42,43]. However, a recent study in gerbil models suggested that carotenoids could exacerbate the vitamin A toxicity of preformed vitamin A when consumed regularly [44]. This recent finding raises the need to better understand the role of the different forms of pro-vitamin A carotenoids in vitamin A toxicity, as well as the need to better manage the combination of interventions to ensure safety for beneficiaries.

Nevertheless, red palm oil is traditionally consumed in both urban and rural areas, while its consumption has decreased due to the availability of refined palm oil. Thus, its use for the substitution of only a part of refined oil could be relevant to ensure coverage of vitamin A without the risks of overconsumption or toxicity.

In the scenarios presented here, ULs were also exceeded for iron, niacin and folate for the scenarios entailing pregnant or lactating women; this was due to the introduction of a voluntary fortified biscuit into the diet. Indeed, the consumption of only one portion per day provides UL levels of iron, niacin and folate. For folate and iron, the ULs were already exceeded with the addition of iron and folic acid supplementation to the basic diet in the Benin urban setting. Consumption of high doses of iron was shown to induce gastrointestinal side effects [45], but the relationship between iron intake and the risk for infectious diseases (e.g., malaria [46] and tuberculosis [47,48]), cardiovascular disease or cancer is less clear [49]. While there is no evidence of toxicity of natural folate in food, folic acid from supplementation or fortification could hamper the diagnosis of pernicious anemia [50]. The risk for niacin overconsumption mainly lies in hepatotoxicity and dermatologic manifestations [5]. Of note, in the absence of the consumption of this biscuit, RNI was not reached for niacin in Benin and Ghana.

Finally, iron was the micronutrient for which the coverage of the needs of non-pregnant women was the most difficult to reach in the different scenarios of this study. Outside supplementation programs, and in the absence of the consumption of voluntary-fortified foods, a high risk for iron deficiency was present in all the scenarios. While, in some cases, the UL set by IOM for iron was reached, it is important to keep in mind that this UL is based on the observation of gastrointestinal side effects of acute high doses, and that the WHO has not set a UL for iron. Thus, more studies are needed to better define the UL for iron, particularly in populations with diets with low iron bioavailability and in the LMICs context, as well as when taking into account the potential impact of iron on intestinal microbiota [51]. Finally, this study also highlights that more actions are needed to better cover the iron needs of women of reproductive age, particularly during the non-pregnancy period, to better ensure adequate iron status.

### 4.5. Strenghts and Limits of the Results

The strength of this study was the use of scenarios based on interventions currently implemented in countries and food menus built on both expert knowledge and food consumption surveys, thus giving us the most credible estimation of the contribution of each intervention to the coverage of women’s needs. A limitation of this study was its consideration of only the main potential scenarios when different food consumption habits might be based on access and availability.

## 5. Conclusions

The scenarios designed in this study were based on daily intakes of micronutrients, with the hypothesis that individuals effectively received interventions targeted for the population they belonged to. Thus, our simulations show different scenarios of how interventions could be combined to improve efficiency in improving the micronutrient status of WRA while avoiding the risk of overconsumption. Fortification programs appear to be most important for ensuring the coverage of micronutrient needs because very few micronutrient needs were covered by the diets themselves. Coordination of the implementation of nutritional programs, including fortification, biofortification, supplementation and the promotion of nutrient-rich food associated with a strict control of the quality of fortified food, is warranted to reduce the high prevalence of micronutrient deficiency in both countries while limiting the risk of overconsumption. This study also reiterates the necessity of screening the needs of populations before the implementation of interventions, as well as the regular evaluation of the impact of these interventions on individuals’ nutritional status.

## Figures and Tables

**Table 1 nutrients-13-02286-t001:** Combination of interventions included in the scenarios proposed for non-pregnant and non-lactating women (NPNL), and pregnant or lactating women (P/L), in urban (U) and rural (R) settings.

	Benin				Ghana			
	NPNL		P/L		NPNL		P/L	
	U	R	U	R	U	R	U	R
Basic diets	X	X	X	X	X	X	X	X
Fortification								
Salt	X	X	X	X			X	X
Oil	X	X	X	X	X	X	X	X
Wheat flour	X		X		X		X	
Biofortification								
Maize					X	X	X	X
Sweet potatoes					X	X	X	X
Supplementation								
IFA			X	X			X	X
High-dose Vit A							X	X
MN-rich food								
Red palm oil	X	X	X	X	X	X	X	X
Fried tilapia	X	X			X	X		
Mango	X	X	X	X	X	X	X	X
Fortified biscuit			X	X			X	X

**Table 2 nutrients-13-02286-t002:** Mandatory micronutrient interventions for women in Benin and Ghana.

		Benin		Ghana	
		Description	Start Year	Description	Start Year
Fortification	Salt iodation	Iodation of salt with 30–40 mg potassium iodate/kg salt	2009	Iodation of salt with 15 ppm of potassium or calcium or sodium iodate per kg salt at household level	1996
	Vegetable oil: vitamin A	retinol palmitate 50 UI/g	2012	10 mg retinol palmitate/kg	2006
	Wheat flour: iron, folic acid, zinc and B vitamins	Wheat Flour with ferrous fumarate (60 g/t); zinc oxide (55 g/t); thiamine (2.79 g/t); riboflavin (2.79 g/t); Niacin (36.18 g/t); pyridoxin: (3.13 g/t); cobalamin (0.02 g/t); folic acid (2.6 g/t)	2012	Cyanocobalamin, 0.01 mg/kg; folic acid, 2.08 mg/kg; ferous fumarate, 58.50 mg/kg; niacinamide, 59.00 mg/kg; Riboflavin, 4.50 mg/kg; thiamine mononitrate, 8.40 mg/kg; retinyl palmitate, 2.00 mg/kg; zinc oxide, 28.30 mg/kg	2010
Biofortification	Orange sweet potatoes: vitamin A	-		Crop released-Variety Cri-Bohye with 38–72 ppm beta-carotene	2009
	Maize: vitamin A	Crop being evaluated	2017	Crop-released variety CSIR-CRI Honampa with at least 6 ppm pro-vitA	2012
	Cassava: vitamin A	Crop being evaluated	2017	Crop being evaluated	2016
	Pearl millet: iron	Crop being evaluated	2017	Crop being evaluated	2017
Supplementation	Iron and folic acid supplementation for pregnant women	IFA program: 60 mg iron and 400 µg folic acid during prenatal consultations	2006	IFA program: 60 mg iron and 400 µg folic acid during prenatal consultations and women with a child between 1 and 5 months	2006
	Vitamin A supplementation post-partum	-		Vitamin A supplementation post-partum: one dose of 200,000 UI	

**Table 3 nutrients-13-02286-t003:** Micronutrient intakes from diet and interventions and coverage of recommendations for non-pregnant, non-lactating women in Benin and Ghana in urban (U) and rural (R) settings. Shaded cells indicate when recommendations or limits are reached.

		Non-Pregnant Women (Benin)												Non-Pregnant Women (Ghana)											
		Fe (mg)		Zn (mg)		Vit. A ** (µg RE)		Niacin (mg)		Folate (µg)		Iodine (µg)		Fe (mg)		Zn (mg)		Vit. A ** (µg RE)		Niacin (mg)		Folate (µg)		Iodine (µg)	
		U	R	U	R	U	R	U	R	U	R	U	R	U	R	U	R	U	R	U	R	U	R	U	R
Diet		14.2	5.0	8.0	2.5	285	146	9.9	3,9	372	134	0.8	0	10.3	5.8	5.7	3.6	249	62	9.4	9.1	188	91	0.3	0
	**% EAR ***	48	17	196	61	80	41	90	35	116	42	1	0	35	20	139	87	70	17	85	83	59	28	0	0
	**% RNI ***	24	9	164	51	57	29	70	28	93	33	1	0	18	10	116	73	50	12	67	65	47	23	0	0
	**% UL ***	32	11	18	6	10	5	28	11	37	13	0	0	23	13	13	8	8	2	27	26	19	9	0	0
Diet + Mandatory fortification		21.3	5.0	14.5	2.5	1934	804	14.1	3.9	687	134	281	354.2	13.9	9.6	7.4	5.4	695	441	13.0	12.9	317	225	107	41
	**% EAR**	72	17	354	61	542	225	128	35	215	42	263	331	47	33	181	132	195	124	118	117	99	70	100	38
	**% RNI**	36	9	296	51	387	161	101	28	172	33	187	236	24	16	151	110	139	88	93	92	79	56	71	27
	**% UL**	47	11	32	6	64	27	40	11	69	13	26	32	31	21	16	12	23	15	37	37	32	23	10	4
Diet+ fortification+ Biofortification														13.3	21.1	7.3	12.4	938	781	10.9	31.2	304	392	107	41
														45	72	178	302	263	219	99	284	95	122	100	38
														23	36	149	252	188	156	78	223	76	98	71	27
														29	47	16	27	31	26	31	89	30	39	10	4
All + MN *** rich food except red palm oil		23.7	7.2	15.7	3.5	2144	1015	17.5	7.1	733	177	281	354.2	15.5	11.5	8.3	6.2	2678	735	14.1	16.1	347	258	107	41
	**% EAR**	81	24	383	85	600	284	159	65	229	55	263	331	53	39	202	151	750	206	128	146	109	81	100	38
	**% RNI**	40	12	321	71	429	203	125	51	183	44	187	236	26	20	169	126	536	147	101	115	87	65	71	27
	**% UL**	53	16	35	8	71	34	50	20	73	18	26	32	34	26	18	14	89	24	40	46	35	26	10	4
All+ fortification except oil fortification		23.7	7.3	15.5	3.5	6777	2866	17.4	7.1	730	177.4	281	354	15.4	11.6	8.3	6.1	1148	2278	14.1	10.2	347	254	107	41
	**% EAR**	80	25	378	85	1898	803	158	65	228	55	263	331	52	39	202	149	322	638	128	92	109	79	100	38
	**% RNI**	40	12	316	71	1355	573	124	51	183	44	187	236	26	20	169	125	230	456	101	73	87	63	71	27
	**% UL**	53	16	34	8	226	96	50	20	73	18	26	32	34	26	18	14	38	76	40	29	35	25	10	234

* % of the EAR, RNI or UL covered by the diet and interventions. EAR: estimated average requirements; RNI: recommended nutrient intakes; UL: upper limits; ** Vit A: vitamin A; *** MN: micronutrients.

**Table 4 nutrients-13-02286-t004:** Micronutrient intakes from diet and interventions and coverage of recommendations for pregnant or lactating women in Benin and Ghana in urban (U) and rural (R) settings. Shaded cells indicate when recommendations or limits are reached.

		Pregnant or Lactating Women (Benin)												Pregnant or Lactating Women (Ghana)											
		Fe (mg)		Zn (mg)		Vit. A ** (µg RE)		Niacin (mg)		Folate (µg)		Iodine (µg)		Fe (mg)		Zn (mg)		Vit. A ** (µg RE)		Niacin (mg)		Folate (µg)		Iodine (µg)	
		U	R	U	R	U	R	U	R	U	R	U	R	U	R	U	R	U	R	U	R	U	R	U	R
Diet		14.9	15.1	8.4	5.4	334	1036	9.9	5.9	416	150	20.8	4.7	7.1	22	5.2	6	324	1663	5.1	3.4	176	71	57.1	9.5
	**% EAR ***	37	38	145	93	58	181	70	42	87	31	15	3	18	54	89	111	57	291	36	24	37	15	40	24
	**% RNI**	50	50	120	77	42	129	55	33	69	25	10	2	24	72	74	92	41	208	28	19	29	12	29	17
	**% UL**	33	34	0	0	11	35	28	17	42	15	1	0	16	48	0	0	11	55	14	10	18	7	2	1
Diet + fortification		22.7	15.1	15.5	5.4	1544	2023	14.5	5.9	749	150	301	568.2	10.9	21.5	7.0	6.4	725	1663	8.8	3.4	310	71	103.3	34.6
	**% EAR**	57	38	267	93	270	354	104	42	156	31	210	397	27	54	121	111	127	291	63	24	64	15	72	24
	**% RNI**	76	50	221	77	193	253	81	33	125	25	150	284	36	72	100	92	91	208	49	19	52	12	52	17
	**% UL**	50	34	0	0	51	67	42	17	75	15	12	22	24	48	0	0	24	55	25	10	31	7	4	1
Diet+ fortification+ supplementation ^§^		82.6	75.1	15.5	5.4	1544	2023	14.5	5.9	1149	550	301	568.2	70.9	81.5	7.0	6.4	60,785	61,723	8.8	3.4	710	471	103.3	34.6
	**% EAR**	207	188	267	93	270	354	104	42	239	115	210	397	177	204	121	111	10,645	10,810	63	24	148	98	72	24
	**% RNI**	275	250	221	77	193	253	81	33	192	92	150	284	236	272	100	92	7598	7715	49	19	118	78	52	17
	**% UL**	184	167	0	0	51	67	42	17	115	55	12	22	157	181	0	0	2026	2057	25	10	71	47	4	1
Diet + fortification + IFA sup + biofortification														70.9	81.7	7.1	6.6	913	1445	9.1	3.6	310	442	103.3	44.0
	**% EAR**													177	204	123	113	160	253	65	26	64	92	72	31
	**% RNI**													236	272	102	94	114	181	51	20	52	74	52	22
	**% UL**													157	182	0	0	30	48	26	10	31	44	4	2
All + MN *** rich food except red palm oil and VAS		221.3	213.8	21.5	11.4	2433	2913	49.9	41.3	2511	1912	301	568.0	210	220	13	13	1803	2335	45	39	2071	1804	103.3	44.0
	**% EAR**	553	535	370	197	426	510	356	295	523	398	210	397	524	551	226	217	316	409	318	278	432	376	72	31
	**% RNI**	738	713	19	163	304	364	277	229	418	319	150	284	699	735	188	179	225	292	247	217	345	301	52	22
	**% UL**	492	475	10	0	81	97	143	118	251	191	12	22	466	490	0	0	60	78	127	111	207	180	4	2
All+ fortification except oil fortification and VAS		222	213.8	21.5	11.4	6571	4881	49.9	41.3	2511	1912	300.9	568.0	210	220	14	13	3183	2335	45	39	2078	1804	103.3	44.0
	**% EAR**	554	535	370	197	1151	855	356	295	523	398	210	397	526	551	233	217	557	409	322	278	433	376	72	31
	**% RNI**	738	713	307	163	821	610	277	229	418	319	150	284	701	735	193	179	398	292	251	217	346	301	52	22
	**% UL**	492	475	1	0	219	163	143	118	251	191	12	22	467	490	0	0	106	78	129	111	208	180	4	2

* EAR: estimated average requirements; RNI: recommended nutrient intakes; UL: upper limits; ** Vit A: vitamin A; *** MN: micronutrients; ^§^ high-dose vitamin A supplementation post-partum for lactating women only.

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
