# Peer review of "Comparison of Micronutrient Intervention Strategies in Ghana and Benin to Cover Micronutrient Needs: Simulation of Bene-Fits and Risks in Women of Reproductive Age"

_nutrients, 2021, doi:10.3390/nu13072286_

Round 1

Reviewer 1 Report

The major goal of this study is to evaluate the benefits and potential risks of different micronutrient intervention programs (supported by international organizations) already ongoing or in process of planing with the aim to reduce the rate of micronutrient depletion in Ghana and Benin. The manuscript is generally written in an aceptable style; the idea of the study is well taken and the study design seems to allow reliable and important conclusions. However, methods description, data presentation, and interpretation can be improved.

  1. The study is a simulation (modelling) of predefined scenarios based on already available data (nutrition behaviour, food composition tables, information on intervention programs etc.) in the countries selected. Consequently, the title should be changed better reflecting what was done (suggestion: „Comparison of micronutrient intervention strategies in Ghana and Benin to cover micronutrient needs: simulation of benefits and risks in the female population“). In addition, such a title includes all aspects of the model, not only risk assessment.
  2. This chapter is too long. All aspects concerning RDAs etc. are not related to the primary goal of the study (modeling!) and should be deleted; important information (e.g., RDA reference values to be reached) are given in the Methods section.

The rationale of the study must be sharpened (e.g., results can/will be used to optimize intervention programs).

  1. Materials and Methods. The four scenarios investigated should be clearly described, e.g. in a separate Table. Presently, the reader must seek this information; Table 1 is of no help in this respect.

It might be also worth not only to mention that „typical daily food consumption patterns were chosen as basic dietary patterns“ and to cite references but to give a short description how this diet is composed (e.g., main food items). It will then be easier to follow the results of „basic“ micronutrient intakes and the intervention methods suggested.   

  1. In Tables 2 and 3, for the first time the four (or five?) scenarios under investigation are at least listed (diet + fortification, diet + fortification + supplementation etc….). This information should also be part of Methods section (see point 3).
  2. Again, the Discussion should clearly focus on a comparison of the different scenarios finally allowing conclusions and recommendations for future activities. Many aspects are already considered but in a sometimes „confusing“ order.       

Reviewer 2 Report

Coverage of Women Micronutrient Needs in Ghana and Benin: Role of Nutritional Interventions and Risk of Excess. Dass et al ,2021

An interesting and important paper looking at the likely and possible micronutrient intakes in women of reproductive age, and the likelihood of under or over consumption of some micronutrients.

It would have been helpful to have line numbers on the pages for pointing out where errors are / suggestions relate to. Also page numbers are not correct after the main tables, and seem to start again. I refer to page numbers at the top of the PDF tab and not on the page.

The abstract is well written and clear, and the background is also well written, summarising the background and justifying the need for the study.

Methods:

  • End of page 2, last paragraph: ‘fails’ should be ‘falls’ under diversification. Perhaps reword this sentence (first on the last paragraph) starting: While not a mandatory program…
  • Diet patterns: second sentence - should this read ‘consultation’ rather than ‘concertation’?
  • There are lots of sources given for how the example menus were devised, but I’m still not clear how the menus and weights were decided. Were the portion sizes from weighted records and did these come from one source or are they an average of different sources. How did you come to a decision if the weight and foods selected were based on several surveys? More precise referencing might help. For example is there specific reference that says urban women eat 6 meals a day and rural women eat 4? Or is this just from local knowledge?

Results:

  • Table 2 – consistency of capital letters for Benin / Ghana in title and in the footnote RNI doesn’t stand for Recommended Daily Allowance.
  • I think it could be clearer on the tables what the % stands for – this is explained in the text but I think it needs to be stated on the table for clarity.
  • Table 3: the title needs to be on the same page as the table. And why is there shading on Table 3 and not table 2? I think the shading of those values that exceed the UL is useful for both tables.

Discussion:

Overall this was an informative discussion on what needs to be done to balance coverage of MN needs whilst avoiding over-consumption. I also liked the consideration of the issues of applying the UL to potentially deficient populations. There are some minor corrections to the English required in places. For example:

 - page 11: top paragraph – allowed ‘coverage’ of iron needs instead of : ‘However, the fortified biscuits were the only intervention that allowed to cover iron needs of WRA’ ; and remove ‘a’ before clear legislation.

- Bio-fortification – ‘tested’ and not ‘testing’

 - page 13: 3rd paragraph ‘ Indeed, the iron, niacin and folate levels reported for this product reached the UL for a consumption of already one portion by day.’ Reword as the last part of the sentence doesn’t make sense.

- References are needed to support some of the statements on page 13: induce the GI effects and relationship to diseases and pernicious anaemia.

-I think there should also be a short section on the limitations to the paper – the need to generalise eating habits and quantities to provide the scenarios, and the limitations of using DRVs for groups of women whose requirements differ such as pregnant and lactating women together in one group.

-The paper ends with a balanced but strong conclusion

References: – there are some minor amendments to the reference list needed regarding consistent use of capital letters and shortened or full length journal names.

  1. Reference 10 – vitamin A and Journal of Nutrition

Reference 11 Benin

Reference 41 WHO and vitamin A
